# Re-evaluating Evaluation

**David Balduzzi**[*]    **Karl Tuyls**[*]    **Julien Perolat**[*]    **Thore Graepel**[*]

## Abstract

*"What we observe is not nature itself, but nature exposed to our method of questioning."* – Werner Heisenberg

Progress in machine learning is measured by careful evaluation on problems of outstanding common interest. However, the proliferation of benchmark suites and environments, adversarial attacks, and other complications has diluted the basic evaluation model by overwhelming researchers with choices. Deliberate or accidental cherry picking is increasingly likely, and designing well-balanced evaluation suites requires increasing effort. In this paper we take a step back and propose *Nash averaging*. The approach builds on a detailed analysis of the *algebraic structure of evaluation* in two basic scenarios: agent-vs-agent and agent-vs-task. The key strength of Nash averaging is that it automatically adapts to redundancies in evaluation data, so that results are not biased by the incorporation of easy tasks or weak agents. Nash averaging thus encourages maximally inclusive evaluation – since there is no harm (computational cost aside) from including all available tasks and agents.

## 1  Introduction

Evaluation is a key driver of progress in machine learning, with e.g. ImageNet [1] and the Arcade Learning Environment [2] enabling subsequent breakthroughs in supervised and reinforcement learning [3, 4]. However, developing evaluations has received little *systematic* attention compared to developing algorithms. Immense amounts of compute is continually expended smashing algorithms and tasks together – *but the results are almost never used to **evaluate and optimize evaluations***. In a striking asymmetry, results are almost exclusively applied to evaluate and optimize algorithms.

The classic train-and-test paradigm on common datasets, which has served the community well [5], is reaching its limits. Three examples suffice. Adversarial attacks have complicated evaluation, raising questions about which attacks to test against [6–9]. Training agents far beyond human performance with self-play means they can only really be evaluated against each other [10, 11]. The desire to build increasingly general-purpose agents has led to a proliferation of environments: Mujoco, DM Lab, Open AI Gym, Psychlab and others [12–15].

In this paper we pause to ask, and partially answer, some basic questions about evaluation: **Q1.** Do tasks test what we think they test? **Q2.** When is a task redundant? **Q3.** Which tasks (and agents) matter the most? **Q4.** How should evaluations be evaluated?

We consider two scenarios: *agent vs task* (AvT), where algorithms are evaluated on suites of datasets or environments; and *agent vs agent* (AvA), where agents compete directly as in Go and Starcraft. Our goal is to treat tasks and agents symmetrically – with a view towards, ultimately, co-optimizing agents and evaluations. From this perspective AvA, where the task is (beating) another agent, is especially interesting. Performance in AvA is often quantified using Elo ratings [16] or the closely related TrueSkill [17]. There are two main problems with Elo. Firstly, Elo bakes-in the assumption that relative skill is transitive; but Elo is meaningless – it has no predictive power – in cyclic games

---

[*]DeepMind. Email: { dbalduzzi | karltuyls | perolat | thore }@google.com

like rock-paper-scissors. Intransitivity has been linked to biodiversity in ecology, and may be useful when evolving populations of agents [18–21]. Secondly, an agent's Elo rating can be inflated by instantiating many copies of an agent it beats (or conversely). This can cause problems when Elo guides hyper-optimization methods like population-based training [22]. Similarly, the most important decision when constructing a task-suite is which tasks to include. It is easy, and all too common, to bias task-suites in favor of particular agents or algorithms.

## 1.1 Overview

Section 2 presents background information on Elo and tools for working with antisymmetric matrices, such as the Schur decomposition and combinatorial Hodge theory. A major theme underlying the paper is that the fundamental algebraic structure of tournaments and evaluation is antisymmetric [23]. Techniques specific to antisymmetric matrices are less familiar to the machine learning community than approaches like PCA that apply to symmetric matrices and are typically correlation-based.

Section 3 presents a unified approach to representing evaluation data, where agents and tasks are treated symmetrically. A basic application of the approach results in our first contribution: a **multidimensional Elo rating** (mElo) that handles cyclic interactions. We also sketch how the Schur decomposition can uncover latent skills and tasks, providing a partial answer to **Q1**. We illustrate mElo on the domain of training an AlphaGo agent [24].

The second contribution of the paper is **Nash averaging**, an evaluation method that is **invariant** to redundant tasks and agents, see section 4. The basic idea is to play a meta-game on evaluation data [25]. The meta-game has a unique maximum entropy Nash equilibrium. The key insight of the paper is that the maxent Nash adapts automatically to the presence of redundant tasks and agents. The maxent Nash distribution thus provides a principled answer to **Q2** and **Q3**: which tasks and agents do and do not matter is determined by a meta-game. Finally, expected difficulty of tasks under the Nash distribution on agents yields a partial answer to **Q4**. The paper concludes by taking a second look at the performance of agents on Atari. We find that, under Nash averaging, human performance ties with the best agents, suggesting better-than-human performance has not yet been achieved.

## 1.2 Related work

Legg and Hutter developed a definition of intelligence which, informally, states "intelligence measures an agent's ability to achieve goals in a wide range of environments" [26,27]. Formally, they consider all computable tasks weighted by algorithmic complexity [28–30]. Besides being incomputable, the distribution places (perhaps overly) heavy weight on the simplest tasks.

A comprehensive study of performance metrics for machine learning and AI can be found in [31–35]. There is a long history of psychometric evaluation in humans, some of which has been applied in artificial intelligence [36–38]. Bradley-Terry models provide a general framework for pairwise comparison [39]. Researchers have recently taken a second look at the arcade learning environment [2] and introduced new performance metrics [40]. However, the approach is quite particular. Recent work using agents to evaluate games has somewhat overlapping motivation with this paper [41–45]. Item response theory is an alternative, and likely complementary, approach to using agents to evaluate tasks [46] that has recently been applied to study the performance of agents on the Arcade Learning Environment [47].

Our approach draws heavily on work applying combinatorial Hodge theory to statistical ranking [48] and game theory [49–51]. We also draw on empirical game theory [52,53], by using a meta-game to "evaluate evaluations", see section 4. Empirical game theory has been applied to domains like poker and continuous double auctions, and has recently been extended to asymmetric games [54–58]. *von Neumann winners* in the dueling bandit setting and NE-regret are related to Nash averaging [59–62].

## 2 Preliminaries

**Notation.** Vectors are column vectors. $\mathbf{0}$ and $\mathbf{1}$ denote the constant vectors of zeros and ones respectively. We sometimes use subscripts to indicate dimensions of vectors and matrices, e.g. $\mathbf{r}_{n \times 1}$ or $\mathbf{S}_{m \times n}$ and sometimes their entries, e.g. $\mathbf{r}_i$ or $\mathbf{S}_{ij}$; no confusion should result. The unit vector with a 1 in coordinate $i$ is $\mathbf{e}_i$. Proofs and code are in the appendix.

## 2.1 The Elo rating system

Suppose $n$ agents play a series of pairwise matches against each other. Elo assigns a rating $r_i$ to each player $i \in [n]$ based on their wins and losses, which we represent as an $n$-vector $\mathbf{r}$. The predicted probability of $i$ beating $j$ given their Elo ratings is

$$\hat{p}_{ij} := \frac{10^{r_i/400}}{10^{r_i/400} + 10^{r_j/400}} = \sigma(\alpha r_i - \alpha r_j), \quad \text{where} \quad \sigma(x) = \frac{1}{1 + e^{-x}} \quad \text{and} \quad \alpha = \frac{\log(10)}{400}.$$

The constant $\alpha$ is not important in what follows, so we pretend $\alpha = 1$. Observe that only the *difference* between Elo ratings affects win-loss predictions. We therefore impose that Elo ratings sum to zero, $\mathbf{r}^\mathsf{T}\mathbf{1} = 0$, without loss of generality. Define the loss,

$$\ell_{\text{Elo}}(p_{ij}, \hat{p}_{ij}) = -p_{ij} \log \hat{p}_{ij} - (1 - p_{ij}) \log(1 - \hat{p}_{ij}), \quad \text{where} \quad \hat{p}_{ij} = \sigma(r_i - r_j)$$

and $p_{ij}$ is the true probability of $i$ beating $j$. Suppose the $t^{\text{th}}$ match pits player $i$ against $j$, with outcome $S_{ij}^t = 1$ if $i$ wins and $S_{ij}^t = 0$ if $i$ loses. Online gradient descent on $\ell_{\text{Elo}}$ obtains

$$r_i^{t+1} \leftarrow r_i^t - \eta \cdot \nabla_{r_i} \ell_{\text{Elo}}(S_{ij}^t, \hat{p}_{ij}^t) = r_i^t + \eta \cdot (S_{ij}^t - \hat{p}_{ij}^t).$$

Choosing learning rate $\eta = 16$ or $32$ recovers the updates introduced by Arpad Elo in [16].

The win-loss probabilities predicted by Elo ratings can fail in simple cases. For example, rock, paper and scissors will all receive the same Elo ratings. Elo's predictions are $\hat{p}_{ij} = \frac{1}{2}$ for all $i, j$ – and so Elo has no predictive power for any given pair of players (e.g. paper beats rock with $p = 1$).

**What are the Elo update's fixed points?** Suppose we batch matches to obtain empirical estimates of the probability of player $i$ beating $j$: $\bar{p}_{ij} = \sum_n \frac{S_{ij}^n}{N_{ij}}$. As the number of matches approaches infinity, the empirical estimates approach the true probabilities $p_{ij}$.

**Proposition 1.** *Elo ratings are at a stationary point under batch updates iff the matrices of empirical probabilities and predicted probabilities have the same row-sums (or, equivalently the same column-sums):*

$$\nabla_{r_i}\left[\sum_j \ell_{Elo}(\bar{p}_{ij}, \hat{p}_{ij})\right] = 0 \ \forall i \quad \text{iff} \quad \sum_j \bar{p}_{ij} = \sum_j \hat{p}_{ij} \ \forall i.$$

Many different win-loss probability matrices result in identical Elo ratings. The situation is analogous to how many different joint probability distributions can have the same marginals. We return to this topic in section 3.1.

## 2.2 Antisymmetric matrices

We recall some basic facts about antisymmetric matrices. Matrix $\mathbf{A}$ is antisymmetric if $\mathbf{A} + \mathbf{A}^\mathsf{T} = 0$. Antisymmetric matrices have even rank and *imaginary* eigenvalues $\{\pm i\lambda_j\}_{j=1}^{\text{rank}(\mathbf{A})/2}$. Any antisymmetric matrix $\mathbf{A}$ admits a real **Schur decomposition**:

$$\mathbf{A}_{n \times n} = \mathbf{Q}_{n \times n} \cdot \mathbf{\Lambda}_{n \times n} \cdot \mathbf{Q}_{n \times n}^\mathsf{T},$$

where $\mathbf{Q}$ is orthogonal and $\mathbf{\Lambda}$ consists of zeros except for $(2 \times 2)$ diagonal-blocks of the form:

$$\mathbf{\Lambda} = \begin{pmatrix} 0 & \lambda_j \\ -\lambda_j & 0 \end{pmatrix}.$$

The entries of $\mathbf{\Lambda}$ are *real* numbers, found by multiplying the eigenvalues of $\mathbf{A}$ by $i = \sqrt{-1}$.

**Proposition 2.** *Given matrix $\mathbf{S}_{m \times n}$ with rank $r$ and singular value decomposition $\mathbf{U}_{m \times r} \mathbf{D}_{r \times r} \mathbf{V}_{r \times n}^\mathsf{T}$. Construct antisymmetric matrix*

$$\mathbf{A}_{(m+n) \times (m+n)} = \begin{pmatrix} \mathbf{0}_{m \times m} & \mathbf{S}_{m \times n} \\ -\mathbf{S}_{n \times m}^\mathsf{T} & \mathbf{0}_{n \times n} \end{pmatrix}.$$

*Then the thin Schur decomposition of $\mathbf{A}$ is $\mathbf{Q}_{(m+n) \times 2r} \mathbf{\Lambda}_{2r \times 2r} \mathbf{Q}_{2r \times (m+n)}^\mathsf{T}$ where the nonzero pairs in $\mathbf{\Lambda}_{2r \times 2r}$ are $\pm$ the singular values in $\mathbf{D}_{r \times r}$ and*

$$\mathbf{Q}_{(m+n) \times 2r} = \begin{pmatrix} -\mathbf{u}_1 & \mathbf{0}_{m \times 1} & \cdots & -\mathbf{u}_r & \mathbf{0}_{m \times 1} \\ \mathbf{0}_{n \times 1} & \mathbf{v}_1 & \cdots & \mathbf{0}_{n \times 1} & \mathbf{v}_r \end{pmatrix}.$$

**Combinatorial Hodge theory** is developed by analogy with differential geometry, see [48–51]. Consider a fully connected graph with vertex set $[n] = \{1, \ldots, n\}$. Assign a **flow** $\mathbf{A}_{ij} \in \mathbb{R}$ to each edge of the graph. The flow in the opposite direction $ji$ is $\mathbf{A}_{ji} = -\mathbf{A}_{ij}$, so flows are just $(n \times n)$ antisymmetric matrices. The flow on a graph is analogous to a vector field on a manifold.

The combinatorial **gradient** of an $n$-vector $\mathbf{r}$ is the flow: $\mathrm{grad}(\mathbf{r}) := \mathbf{r1}^{\mathsf{T}} - \mathbf{1r}^{\mathsf{T}}$. Flow $\mathbf{A}$ is a **gradient flow** if $\mathbf{A} = \mathrm{grad}(\mathbf{r})$ for some $\mathbf{r}$, or equivalently if $\mathbf{A}_{ij} = \mathbf{r}_i - \mathbf{r}_j$ for all $i, j$. The **divergence** of a flow is the $n$-vector $\mathrm{div}(\mathbf{A}) := \frac{1}{n} \mathbf{A} \cdot \mathbf{1}$. The divergence measures the contribution to the flow of each vertex, considered as a source. The **curl** of a flow is the three-tensor $\mathrm{curl}(\mathbf{A})_{ijk} = \mathbf{A}_{ij} + \mathbf{A}_{jk} - \mathbf{A}_{ik}$. Finally, the **rotation** is $\mathrm{rot}(\mathbf{A})_{ij} = \frac{1}{n} \sum_{k=1}^{n} \mathrm{curl}(\mathbf{A})_{ijk}$.

**Theorem** (Hodge decomposition, [48]). *(i)* $\mathrm{div} \circ \mathrm{grad}(\mathbf{r}) = \mathbf{r}$ *for any* $\mathbf{r}$ *satisfying* $\mathbf{r}^{\mathsf{T}}\mathbf{1} = 0$. *(ii)* $\mathrm{div} \circ \mathrm{rot}(\mathbf{A}) = \mathbf{0}_{n \times 1}$ *for any flow* $\mathbf{A}$. *(iii)* $\mathrm{rot} \circ \mathrm{grad}(\mathbf{r}) = \mathbf{0}_{n \times n}$ *for any vector* $\mathbf{r}$. *(iv) The vector space of antisymmetric matrices admits an orthogonal decomposition*

$$\{\textit{flows}\} = \{\textit{antisymmetric matrices}\} = \mathrm{im}(\mathrm{grad}) \oplus \mathrm{im}(\mathrm{rot})$$

*with respect to the standard inner product* $\langle \mathbf{A}, \mathbf{B} \rangle = \sum_{ij} \mathbf{A}_{ij} \mathbf{B}_{ij}$. *Concretely, any antisymmetric matrix decomposes as*

$$\mathbf{A} = \{\textit{transitive component}\} + \{\textit{cyclic component}\} = \mathrm{grad}(\mathbf{r}) + \mathrm{rot}(\mathbf{A}) \quad \textit{where} \quad \mathbf{r} = \mathrm{div}(\mathbf{A}).$$

**Sneak peak.** The divergence recovers Elo ratings or just plain average performance depending on the scenario. The Hodge decomposition separates transitive (captured by averages or Elo) from cyclic interactions (rock-paper-scissors), and explains when Elo ratings make sense. The Schur decomposition is a window into the latent skills and tasks not accounted for by Elo and averages.

## 3 On the algebraic structure of evaluation

The Schur decomposition and combinatorial Hodge theory provide a unified framework for analyzing evaluation data in the AvA and AvT scenarios. In this section we provide some basic tools and present a multidimensional extension of Elo that handles cyclic interactions.

### 3.1 Agents vs agents (AvA)

In AvA, results are collated into a matrix of win-loss probabilities based on relative frequencies. Construct $\mathbf{A} = \mathrm{logit}\,\mathbf{P}$ with $\mathbf{A}_{ij} := \log \frac{p_{ij}}{1 - p_{ij}}$. Matrix $\mathbf{A}$ is antisymmetric since $p_{ij} + p_{ji} = 1$.

**When can Elo correctly predict win-loss probabilities?** The answer is simple in logit space:

**Proposition 3.** *(i) If probabilities* $\mathbf{P}$ *are generated by Elo ratings* $\mathbf{r}$ *then the divergence of its logit is* $\mathbf{r}$*. That is,*

$$\textit{if } p_{ij} = \sigma(r_i - r_j) \; \forall i, j \; \textit{ then } \; \mathrm{div}(\mathrm{logit}\,\mathbf{P}) = \left( \frac{1}{n} \sum_{j=1}^{n} (r_i - r_j) \right)_{i=1}^{n} = \mathbf{r}.$$

*(ii) There is an Elo rating that generates probabilities* $\mathbf{P}$ *iff* $\mathrm{curl}(\mathrm{logit}\,\mathbf{P}) = 0$. *Equivalently, iff* $\log \frac{p_{ij}}{p_{ji}} + \log \frac{p_{jk}}{p_{kj}} + \log \frac{p_{ki}}{p_{ik}} = 0$ *for all* $i, j, k$.

Elo is, essentially, a uniform average in logit space. Elo's predictive failures are due to the cyclic component $\tilde{\mathbf{A}} := \mathrm{rot}(\mathrm{logit}\,\mathbf{P})$ that uniform averaging ignores.

**Multidimensional Elo (mElo$_{2k}$).** Elo ratings bake-in the assumption that relative skill is transitive. However, there is no single dominant strategy in games like rock-paper-scissors or (arguably) StarCraft. Rating systems that can handle intransitive abilities are therefore necessary. An obvious approach is to learn a feature vector $\mathbf{w}$ and a rating vector $\mathbf{r}_i$ per player, and predict $\hat{p}_{ij} = \sigma(\mathbf{r}_i^{\mathsf{T}}\mathbf{w} - \mathbf{r}_j^{\mathsf{T}}\mathbf{w})$. Unfortunately, this reduces to the standard Elo rating since $\mathbf{r}_i^{\mathsf{T}}\mathbf{w}$ is a scalar.

Handling intransitive abilities requires learning an approximation to the cyclic component $\tilde{\mathbf{A}}$. Combining the Schur and Hodge decompositions allows to construct low-rank approximations that extend Elo. Note, antisymmetric matrices have *even* rank. Consider

$$\mathbf{A}_{n \times n} = \mathrm{grad}(\mathbf{r}) + \tilde{\mathbf{A}} \approx \mathrm{grad}(\mathbf{r}) + \mathbf{C}^{\mathsf{T}} \begin{pmatrix} 0 & 1 & \\ -1 & 0 & \\ & & \ddots \end{pmatrix} \mathbf{C} =: \mathrm{grad}(\mathbf{r}) + \mathbf{C}_{n \times 2k}^{\mathsf{T}} \mathbf{\Omega}_{2k \times 2k} \mathbf{C}_{2k \times n}$$

where the rows of $\mathbf{C}$ are orthogonal to each other, to $\mathbf{r}$, and to $\mathbf{1}$. The larger $2k$, the better the approximation. Let $\mathbf{mElo}_{2k}$ assign each player Elo rating $r_i$ and $2k$-dimensional vector $\mathbf{c}_i$. Vanilla Elo uses $2k = 0$. The $\mathrm{mElo}_{2k}$ win-loss prediction is

$$\mathbf{mElo}_{2k}\colon \; \hat{p}_{ij} = \sigma\Big(r_i - r_j + \mathbf{c}_i^{\mathsf{T}} \cdot \mathbf{\Omega}_{2k \times 2k} \cdot \mathbf{c}_j\Big) \text{ where } \mathbf{\Omega}_{2k \times 2k} = \sum_{i=1}^{k}(\mathbf{e}_{2i-1}\mathbf{e}_{2i}^{\mathsf{T}} - \mathbf{e}_{2i}\mathbf{e}_{2i-1}^{\mathsf{T}}).$$

Online updates can be computed by gradient descent, see section E, with orthogonality enforced.

## 3.2 Application: predicting win-loss probabilities in Go

Elo ratings are widely used in Chess and Go. We compared the predictive capabilities of Elo and the simplest extension $\mathrm{mElo}_2$ on eight Go algorithms taken from extended data table 9 in [24]: seven variants of AlphaGo, and Zen. The Frobenius norms and logistic losses are $\|\mathbf{P} - \hat{\mathbf{P}}\|_F = 0.85$ and $\ell_{\log} = 1.41$ for Elo vs the empirical probabilities and $\|\mathbf{P} - \hat{\mathbf{P}}_2\|_F = 0.35$ and $\ell_{\log} = 1.27$ for $\mathrm{mElo}_2$.

To better understand the difference, we zoom in on three algorithms that were observed to interact non-transitively in [58]: $\alpha_v$ with value net, $\alpha_p$ with policy net, and Zen. Elo's win-loss predictions are poor (Table **Elo**: Elo incorrectly predicts both that $\alpha_p$ likely beats $\alpha_v$ and $\alpha_v$ likely beats Zen), whereas $\mathrm{mElo}_2$ (Table **mElo**$_2$) correctly predicts likely winners in all cases (Table **empirical**), with more accurate probabilities:

| Elo | $\alpha_v$ | $\alpha_p$ | Zen |
|---|---|---|---|
| $\alpha_v$ | - | 0.41 | 0.58 |
| $\alpha_p$ | 0.59 | - | 0.67 |
| Zen | 0.42 | 0.33 | - |

| empirical | $\alpha_v$ | $\alpha_p$ | Zen |
|---|---|---|---|
| $\alpha_v$ | - | 0.7 | 0.4 |
| $\alpha_p$ | 0.3 | - | 1.0 |
| Zen | 0.6 | 0.0 | - |

| mElo$_2$ | $\alpha_v$ | $\alpha_p$ | Zen |
|---|---|---|---|
| $\alpha_v$ | - | 0.72 | 0.46 |
| $\alpha_p$ | 0.28 | - | 0.98 |
| Zen | 0.55 | 0.02 | - |

## 3.3 Agents vs tasks (AvT)

In AvT, results are represented as an $(m \times n)$ matrix $\mathbf{S}$: rows are agents, columns are tasks, entries are scores (e.g. accuracy or total reward). Subtract the total mean, so the sum of all entries of $\mathbf{S}$ is zero. We recast both agents and tasks as *players* and construct an antisymmetric $(m+n) \times (m+n)$-matrix. Let $\mathbf{s} = \frac{1}{m}\mathbf{S} \cdot \mathbf{1}_{m \times 1}$ and $\mathbf{d} = -\frac{1}{n}\mathbf{S}^{\mathsf{T}} \cdot \mathbf{1}_{n \times 1}$ be the **average skill of each agent** and the **average difficulty of each task**. Define $\tilde{\mathbf{S}} = \mathbf{S} - (\mathbf{s} \cdot \mathbf{1}^{\mathsf{T}} - \mathbf{1} \cdot \mathbf{d}^{\mathsf{T}})$. Let $\mathbf{r}$ be the concatenation of $\mathbf{s}$ and $\mathbf{d}$. We construct the antisymmetric matrix

$$\mathbf{A}_{(m+n) \times (m+n)} = \mathrm{grad}(\mathbf{r}) + \underbrace{\begin{pmatrix} \mathbf{0}_{m \times m} & \tilde{\mathbf{S}}_{m \times n} \\ -\tilde{\mathbf{S}}_{n \times m}^{\mathsf{T}} & \mathbf{0}_{n \times n} \end{pmatrix}}_{\tilde{\mathbf{A}}} = \begin{pmatrix} \mathrm{grad}(\mathbf{s}) & \mathbf{S}_{m \times n} \\ -\mathbf{S}_{n \times m}^{\mathsf{T}} & \mathrm{grad}(\mathbf{d}) \end{pmatrix}.$$

The top-right block of $\mathbf{A}$ is agent performance on tasks; the bottom-left is task difficulty for agents. The top-left block compares agents by their average skill on tasks; the bottom-right compares tasks by their average difficulty for agents. Average skill and difficulty explain the data if the score of agent $i$ on task $j$ is $\mathbf{S}_{ij} = \mathbf{s}_i - \mathbf{d}_j$, the agent's skill minus the task's difficulty, for all $i, j$. Paralleling proposition 3, averages explain the data, $\mathbf{S} = \mathbf{s}\mathbf{1}^{\mathsf{T}} - \mathbf{1}\mathbf{d}^{\mathsf{T}}$, iff $\mathrm{curl}(\mathbf{A}) = \mathbf{0}$.

The failure of averages to explain performance is encapsulated in $\tilde{\mathbf{S}}$ and $\tilde{\mathbf{A}}$. By proposition 2, the SVD of $\tilde{\mathbf{S}}$ and Schur decomposition of $\tilde{\mathbf{A}}$ are equivalent. If the SVD is $\tilde{\mathbf{S}}_{m \times n} = \mathbf{U}_{m \times r}\mathbf{D}_{r \times r}\mathbf{V}_{r \times n}^{\mathsf{T}}$ then the rows of $\mathbf{U}$ represent the latent abilities exhibited by agents and the rows of $\mathbf{V}$ represent the latent problems posed by tasks.

# 4 Invariant evaluation

Evaluation is often based on metrics like average performance or Elo ratings. Unfortunately, two (or two hundred) tasks or agents that look different may test/exhibit identical skills. Overrepresenting particular tasks or agents introduces biases into averages and Elo – biases that can only be detected *post hoc*. Humans must therefore decide which tasks or agents to retain, to prevent redundant agents or tasks from skewing results. At present, *evaluation is not automatic and does not scale.* To be scalable and automatic, an evaluation method should *always benefit* from including additional agents and tasks. Moreover, it should *adjust automatically and gracefully* to redundant data.

**Definition 1.** *An **evaluation method** maps data to a real-valued function on players (that is, agents or agents and tasks):*

$$\mathcal{E} : \big\{ \text{evaluation data} \big\} = \big\{ \text{antisymmetric matrices} \big\} \rightarrow \Big[ \big\{ \text{players} \big\} \rightarrow \mathbb{R} \Big].$$

**Desired properties.** An evaluation method should be:

P1. *Invariant:* adding redundant copies of an agent or task to the data should make no difference.

P2. *Continuous:* the evaluation method should be robust to small changes in the data.

P3. *Interpretable:* hard to formalize, but the procedure should agree with intuition in basic cases.

Elo and uniform averaging over tasks are examples of evaluation methods that invariance excludes.

## 4.1 Nash averaging

This section presents an evaluation method satisfying properties $P1, P2, P3$. We discuss AvA here, see section D for AvT. Given antisymmetric logit matrix $\mathbf{A}$, define a two-player meta-game with *payoffs* $\mu_1(\mathbf{p}, \mathbf{q}) = \mathbf{p}^\intercal \mathbf{A} \mathbf{q}$ and $\mu_2(\mathbf{p}, \mathbf{q}) = \mathbf{p}^\intercal \mathbf{B} \mathbf{q}$ for the row and column meta-players, where $\mathbf{B} = \mathbf{A}^\intercal$. The game is symmetric because $\mathbf{B} = \mathbf{A}^\intercal$ and zero-sum because $\mathbf{B} = -\mathbf{A}$.

The row and column meta-players pick 'teams' of agents. Their payoff is the expected log-odds of their respective team winning under the joint distribution. If there is a dominant agent that has better than even odds of beating the rest, both players will pick it. In rock-paper-scissors, the only unbeatable-on-average team is the uniform distribution. In general, the value of the game is zero and the Nash equilibria are teams that are unbeatable in expectation.

A problem with Nash equilibria (NE) is that they are not unique, which forces the user to make choices and undermines interpretability [63, 64]. Fortunately, for zero-sum games there is a natural choice of Nash:

**Proposition 4** (maxent NE). *For antisymmetric $\mathbf{A}$ there is a unique symmetric Nash equilibrium $(\mathbf{p}^*, \mathbf{p}^*)$ solving $\max_{\mathbf{p} \in \Delta_n} \min_{\mathbf{q} \in \Delta_n} \mathbf{p}^\intercal \mathbf{A} \mathbf{q}$ with greater entropy than any other Nash equilibrium.*

Maxent Nash is maximally indifferent between players with the same empirical performance.

**Definition 2.** *The **maxent Nash evaluation method** for AvA is*

$$\mathcal{E}_m : \big\{ \text{evaluation data} \big\} = \big\{ \text{antisymmetric matrices} \big\} \xrightarrow{\text{maxent NE}} \Big[ \big\{ \text{players} \big\} \xrightarrow{\text{Nash average}} \mathbb{R} \Big],$$

*where $\mathbf{p}_{\mathbf{A}}^*$ is the **maxent Nash equilibrium** and $\mathbf{n}_{\mathbf{A}} := \mathbf{A} \cdot \mathbf{p}_{\mathbf{A}}^*$ is the **Nash average**.*

Invariance to redundancy is best understood by looking at an example; for details see section C.

*Example* 1 (invariance). Consider two logit matrices, where the second adds a redundant copy of agent $C$ to the first:

| $\mathbf{A}$ | $A$ | $B$ | $C$ |
|---|---|---|---|
| $A$ | 0.0 | 4.6 | -4.6 |
| $B$ | -4.6 | 0.0 | 4.6 |
| $C$ | 4.6 | -4.6 | 0.0 |

and

| $\mathbf{A}'$ | $A$ | $B$ | $C_1$ | $C_2$ |
|---|---|---|---|---|
| $A$ | 0.0 | 4.6 | -4.6 | -4.6 |
| $B$ | -4.6 | 0.0 | 4.6 | 4.6 |
| $C_1$ | 4.6 | -4.6 | 0.0 | 0.0 |
| $C_2$ | 4.6 | -4.6 | 0.0 | 0.0 |

The maxent Nash for $\mathbf{A}$ is $\mathbf{p}_{\mathbf{A}}^* = (\frac{1}{3}, \frac{1}{3}, \frac{1}{3})$. It is easy to check that $(\frac{1}{3}, \frac{1}{3}, \frac{\alpha}{3}, \frac{1-\alpha}{3})$ is Nash for $\mathbf{A}'$ for any $\alpha \in [0, 1]$ and thus the maxent Nash for $\mathbf{A}'$ is $\mathbf{p}_{\mathbf{A}'}^* = (\frac{1}{3}, \frac{1}{3}, \frac{1}{6}, \frac{1}{6})$. Maxent Nash automatically detects the redundant agents $C_1, C_2$ and distributes $C$'s mass over them equally.

Uniform averaging is not invariant to adding redundant agents; concretely $\text{div}(\mathbf{A}) = \mathbf{0}$ whereas $\text{div}(\mathbf{A}') = (-1.15, 1.15, 0, 0)$, falsely suggesting agent $B$ is superior. In contrast, $\mathbf{n}_{\mathbf{A}} = \mathbf{0}_{3 \times 1}$ and $\mathbf{n}_{\mathbf{A}'} = \mathbf{0}_{4 \times 1}$ (the zero-vectors have different sizes because there are different numbers of agents). Nash averaging correctly reports no agent is better than the rest in both cases.

**Theorem 1** (main result for AvA[2]). *The maxent NE has the following properties:*

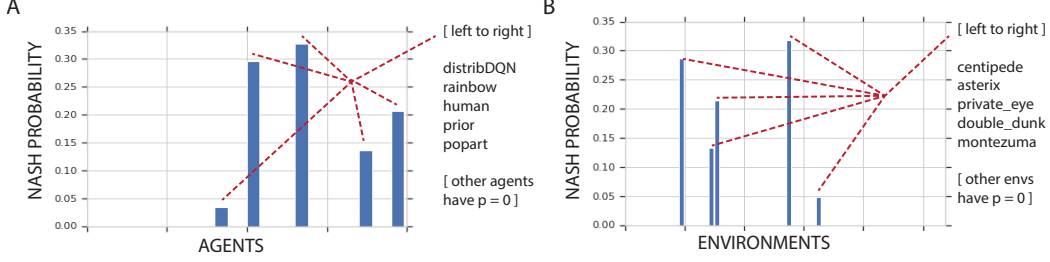

Figure 1: (A) The Nash $\mathbf{p}_a^*$ assigned to agents; (B) the Nash $\mathbf{p}_e^*$ assigned to environments.

**P1. *Invariant:*** *Nash averaging, with respect to the maxent NE, is invariant to redundancies in* $\mathbf{A}$.

**P2. *Continuous:*** *If* $\mathbf{p}^*$ *is a Nash for* $\hat{\mathbf{A}}$ *and* $\epsilon = \|\mathbf{A} - \hat{\mathbf{A}}\|_{max}$ *then* $\mathbf{p}^*$ *is an* $\epsilon$*-Nash for* $\mathbf{A}$.

**P3. *Interpretable: (i)*** *The maxent NE on* $\mathbf{A}$ *is the uniform distribution,* $\mathbf{p}^* = \frac{1}{n}\mathbf{1}$, *iff the meta-game is* cyclic, *i.e.* $\mathrm{div}(\mathbf{A}) = \mathbf{0}$. ***(ii)*** *If the meta-game is* transitive, *i.e.* $\mathbf{A} = \mathrm{grad}(\mathbf{r})$, *then the maxent NE is the uniform distribution on the player(s) with highest rating(s) – there could be a tie.*

See section C for proof and formal definitions. For interpretability, if $\mathbf{A} = \mathrm{grad}(\mathbf{r})$ then the transitive rating is all that matters: Nash averaging measures performance against the best player(s). If $\mathrm{div}(\mathbf{A}) = \mathbf{0}$ then no player is better than any other. Mixed cases cannot be described in closed form.

The continuity property is quite weak: theorem 1.2 shows the *payoff* is continuous: *a team that is unbeatable for* $\hat{\mathbf{A}}$ *is* $\epsilon$*-beatable for nearby* $\mathbf{A}$. Unfortunately, Nash equilibria themselves can jump discontinuously when $\mathbf{A}$ is modified slightly. Perturbed best response converges to a more stable approximation to Nash [65,66] that unfortunately is not invariant.

*Example* 2 (continuity). Consider the cyclic and transitive logit matrices

$$\mathbf{C} = \begin{pmatrix} 0 & 1 & -1 \\ -1 & 0 & 1 \\ 1 & -1 & 0 \end{pmatrix} \quad \text{and} \quad \mathbf{T} = \begin{pmatrix} 0 & 1 & 2 \\ -1 & 0 & 1 \\ -2 & -1 & 0 \end{pmatrix}.$$

The maxent Nash equilibria and Nash averages of $\mathbf{C} + \epsilon\mathbf{T}$ are

$$\mathbf{p}_{\mathbf{C}+\epsilon\mathbf{T}}^* = \begin{cases} \left(\frac{1+\epsilon}{3}, \frac{1-2\epsilon}{3}, \frac{1+\epsilon}{3}\right) & \text{if } 0 \le \epsilon \le \frac{1}{2} \\ (1,0,0) & \text{if } \frac{1}{2} < \epsilon \end{cases} \quad \text{and} \quad \mathbf{n}_{\mathbf{C}+\epsilon\mathbf{T}} = \begin{cases} (0,0,0) & 0 \le \epsilon \le \frac{1}{2} \\ (0,-1-\epsilon, 1-2\epsilon) & \frac{1}{2} < \epsilon \end{cases}$$

The maxent Nash is the uniform distribution over agents in the cyclic case ($\epsilon = 0$), and is concentrated on the first player when it dominates the others ($\epsilon > \frac{1}{2}$). When $0 < \epsilon < \frac{1}{2}$ the optimal team has most mass on the first and last players. Nash jumps discontinuously at $\epsilon = \frac{1}{2}$.

## 4.2 Application: re-evaluation of agents on the Arcade Learning Environment

To illustrate the method, we re-evaluate the performance of agents on Atari [2]. Data is taken from results published in [67–70]. Agents include `rainbow`, dueling networks, `prioritized` replay, `pop-art`, DQN, count-based exploration and baselines like `human`, `random`-action and `no-action`. The 20 agents evaluated on 54 environments are represented by matrix $\mathbf{S}_{20\times54}$. It is necessary to standardize units across environments with quite different reward structures: for each column we subtract the min and divide by the max so scores lie in $[0,1]$.

We introduce a meta-game where row meta-player picks aims to pick the best distribution $\mathbf{p}_a^*$ on agents and column meta-player aims to pick the hardest distribution $\mathbf{p}_e^*$ on environments, see section D for details. We find a Nash equilibrium using an LP-solver; it should be possible to find the maxent Nash using the algorithm in [71,72]. The Nash distributions are shown in figure 1. The supports of the distributions are the 'core agents' and the 'core environments' that form unexploitable teams. See appendix for tables containing all skills and difficulties. panel B.

Figure 2A shows the skill of agents under uniform $\frac{1}{n}\mathbf{S}\cdot\mathbf{1}$ and Nash $\mathbf{S}\cdot\mathbf{p}_e^*$ averaging over environments; panel B shows the difficulty of environments under uniform $-\frac{1}{m}\mathbf{S}^{\mathsf{T}}\cdot\mathbf{1}$ and Nash $-\mathbf{S}^{\mathsf{T}}\cdot\mathbf{p}_a^*$ averaging over agents. There is a tie for top between the agents with non-zero mass – including human. This follows by the indifference principle for Nash equilibria: strategies with support have equal payoff.

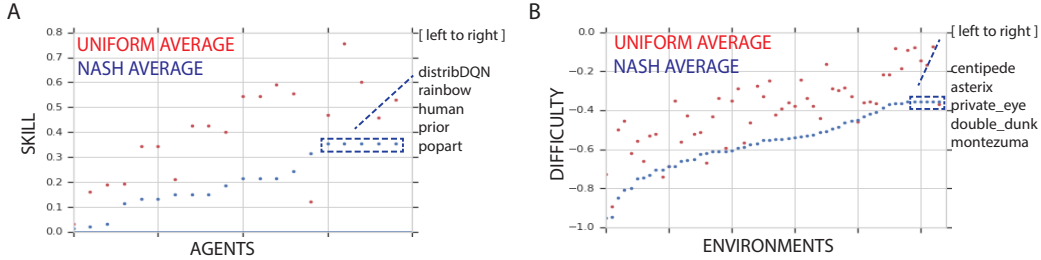

Figure 2: **Comparison of uniform and Nash averages.** (A) Skill of agents by uniform $\frac{1}{n}\mathbf{S} \cdot \mathbf{1}$ and Nash $\mathbf{S} \cdot \mathbf{p}_e^*$ averaging over environments. (B) Difficulty of environments under uniform $-\frac{1}{m}\mathbf{S}^\mathsf{T} \cdot \mathbf{1}$ and Nash $-\mathbf{S}^\mathsf{T} \cdot \mathbf{p}_a^*$ averaging over agents. Agents and environments are sorted by Nash-averages.

Our results suggest that the better-than-human performance observed on the Arcade Learning Environment is because ALE is skewed towards environments that (current) agents do well on, and contains fewer environments testing skills specific to humans. Solving the meta-game automatically finds a distribution on environments that evens out the playing field and, simultaneously, identifies the most important agents and environments.

## 5 Conclusion

A powerful guiding principle when deciding what to measure is to find quantities that are *invariant* to naturally occurring transformations. The determinant is computed over a basis – however, the determinant is *invariant to the choice of basis* since $\det(G^{-1}AG) = \det(A)$ for any invertible matrix $G$. Noether's theorem implies the dynamics of a physical system with symmetries obeys a conservation law. The speed of light is fundamental because it is invariant to the choice of inertial reference frame.

One must have symmetries in mind to talk about invariance. **What are the naturally occurring symmetries in machine learning?** The question admits many answers depending on the context, see e.g. [73–79]. In the context of evaluating agents, that are typically built from neural networks, it is unclear *a priori* whether two seemingly different agents – based on their parameters or hyperparameters – are *actually* different. Further, it is increasingly common that environments and tasks are parameterized – or are learning agents in their own right, see self-play [10,11], adversarial attacks [6–9], and automated curricula [80]. The overwhelming source of symmetry when evaluating learning algorithms is therefore **redundancy**: different agents, networks, algorithms, environments and tasks that do basically the same job.

Nash evaluation computes a distribution on players (agents, or agents and tasks) that automatically adjusts to redundant data. It thus provides an invariant approach to measuring agent-agent and agent-environment interactions. In particular, Nash averaging encourages a maximally inclusive approach to evaluation: computational cost aside, the method should only benefit from including as many tasks and agents as possible. Easy tasks or poorly performing agents will not bias the results. As such Nash averaging is a significant step towards more *objective* evaluation.

Nash averaging is not always the right tool. Firstly, it is only as good as the data: *garbage in, garbage out*. Nash decides which environments are important based on the agents provided to it, and conversely. As a result, the method is blind to differences between environments that do not make a difference to agents and vice versa. Nash-based evaluation is likely to be most effective when applied to a diverse array of agents and environments. Secondly, for good or ill, Nash averaging removes control from the user. One may have good reason to disagree with the distribution chosen by Nash. Finally, Nash is a harsh master. It takes an adversarial perspective and may not be the best approach to, say, constructing automated curricula – although boosting is a related approach that works well [81,82]. It is an open question whether alternate invariant evaluations can be constructed, game-theoretically or otherwise.

**Acknowledgements.** We thank Georg Ostrovski, Pedro Ortega, José Hernández-Orallo and Hado van Hasselt for useful feedback.

## Footnotes

[2]The main result for AvT is analogous, see section D.

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
