[Supplementary Material]

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

# APPENDIX

## A   Invariance: Further motivation

Consider agents A, B, C evaluated on a benchmark suite comprising three tasks:

|         | task 1 | task 2 | task 3 | **average** | **rank** |
|---------|--------|--------|--------|-------------|----------|
| agent A | 89     | 93     | 76     | 86          | $1^{st}$ |
| agent B | 85     | 85     | 85     | 85          | $2^{nd}$ |
| agent C | 79     | 74     | 99     | 84          | $3^{rd}$ |

On average, agent A performs best and agent C performs worst. Consider a second benchmark suite containing an additional fourth task. On the second benchmark suite, agent A performs worst and agent C performs best on average. However, a closer look at the second suite reveals that the additional task is a minor variant of one of the original three tasks.

|         | task 1 | task 2 | task 3a | task 3b | **average** | **rank** |
|---------|--------|--------|---------|---------|-------------|----------|
| agent A | 89     | 93     | 76      | 77      | 84          | $3^{rd}$ |
| agent B | 85     | 85     | 85      | 84      | 85          | $2^{nd}$ |
| agent C | 79     | 74     | 99      | 98      | 88          | $1^{st}$ |

Measuring performance by uniformly averaging over tasks in a benchmark suite is sensitive to the set of tasks that are included in the suite. Including redundant tasks, whether consciously or not, can easily skew average performance in favor of or against particular agents.

The problem becomes more serious when agent performance is measured against other agents, as in games such as Go, Chess or StarCraft. It is easy to manipulate the measured performance of agents by tweaking the composition of the population used to evaluate them. Consider the following example:

|         | agent A | agent B | agent C | **Elo** |
|---------|---------|---------|---------|---------|
| agent A | 0.5     | 0.9     | 0.1     | 0       |
| agent B | 0.1     | 0.5     | 0.9     | 0       |
| agent C | 0.9     | 0.1     | 0.5     | 0       |

The three agents exhibit rock-paper-scissors dynamics; their Elo ratings (normalized to sum to zero) are all zero. However, adding a second copy of agent C decreases the Elo rating of agent A and increases the Elo rating of agent B:

|              | agent A | agent B | agent $C_1$ | agent $C_2$ | **Elo** |
|--------------|---------|---------|-------------|-------------|---------|
| agent A      | 0.5     | 0.9     | 0.1         | 0.1         | -63     |
| agent B      | 0.1     | 0.5     | 0.9         | 0.9         | 63      |
| agent $C_1$  | 0.9     | 0.1     | 0.5         | 0.5         | 0       |
| agent $C_2$  | 0.9     | 0.1     | 0.5         | 0.5         | 0       |

That is, the Elo ratings of agents A and B are easily manipulated by changing the structure of the population.

The examples above suggest it is important to find evaluation metrics that are *invariant* to redundant changes in the population of agents or suite of tasks.

**Related work.** A different notion of measurement invariance has been proposed in the psychometric and consumer research literatures [83]. There, measurement invariance refers to the statistical property that a measurement measures the same construct across a predefined set of groups. For example, whether or not a question in an IQ test is measurement invariant has to do with whether or not the question is interpreted in the same way by individuals with different cultural backgrounds.

# B   Proofs of propositions

**Proof of proposition 1.**

**Proposition 1.** *Batch Elo updates are stable if the matrices of empirical probabilities and predicted probabilities have the same row-sums (or, equivalently the same column-sums):*

$$\nabla_i \sum_j \ell_{Elo}(\bar{p}_{ij}, \hat{p}_{ij}) = 0 \ \forall i \quad \textit{iff} \quad \sum_j \bar{p}_{ij} = \sum_j \hat{p}_{ij} \ \forall i.$$

*Proof.* Player $i$'s weights are updated in one batch after observing win-loss probabilities $\bar{p}_{ij}$ for each player $j = 1, \ldots, n$. Observe that

$$\nabla_i \left[ \sum_{j=1}^n \ell_{\text{Elo}}(\bar{p}_{ij}, \hat{p}_{ij}) \right] = \sum_{j=1}^n (\bar{p}_{ij} - \hat{p}_{ij}).$$

The result follows. $\qquad\qquad\square$

**Proof of proposition 2.**

**Proposition 2.** *Given matrix $\mathbf{S}_{m \times n}$ with rank $r$ and singular value decomposition $\mathbf{UDV}^{\mathsf{T}}$. Construct antisymmetric matrix*

$$\mathbf{A}_{(m+n) \times (m+n)} = \begin{pmatrix} \mathbf{0}_{m \times m} & \mathbf{S}_{m \times n} \\ -\mathbf{S}_{n \times m}^{\mathsf{T}} & \mathbf{0}_{n \times n} \end{pmatrix}.$$

*Then the thin Schur decomposition of $\mathbf{A}$ is $\mathbf{Q}\mathbf{\Lambda}\mathbf{Q}^{\mathsf{T}}$ where the eigenpairs in the $\mathbf{\Lambda}_{2r \times 2r}$ are $\pm$ the singular values in $\mathbf{D}$ and*

$$\mathbf{Q}_{(m+n) \times r} = \begin{pmatrix} -\mathbf{u}_1 & \mathbf{0}_{m \times 1} & \cdots & -\mathbf{u}_r & \mathbf{0}_{m \times 1} \\ \mathbf{0}_{n \times 1} & \mathbf{v}_1 & \cdots & \mathbf{0}_{n \times 1} & \mathbf{v}_r \end{pmatrix}$$

*Proof.* Direct computation; multiply out the matrices. $\qquad\qquad\square$

**Proof of proposition 3.**

**Proposition 3.** *(i) If probabilities $\mathbf{P}$ are generated by Elo ratings $\mathbf{r}$ then the divergence of its logit is $\mathbf{r}$. That is,*

$$\textit{if } p_{ij} = \sigma(r_i - r_j) \ \forall i, j \ \textit{ then } \ \text{div}(\text{logit}\,\mathbf{P}) = \left( \frac{1}{n} \sum_{j=1}^n (r_i - r_j) \right)_{i=1}^n = \mathbf{r}.$$

*(ii) There is an Elo rating that generates probabilities $\mathbf{P}$ iff $\text{curl}(\text{logit}\,\mathbf{P}) = 0$. Alternatively, iff $\log \frac{p_{ij}}{p_{ji}} + \log \frac{p_{jk}}{p_{kj}} + \log \frac{p_{ki}}{p_{ik}} = 0$ for all $i, j, k$.*

*Proof.* For the first claim, apply definitions and recall $\mathbf{r}^{\mathsf{T}}\mathbf{1} = 0$. For the second, apply the Hodge decomposition from section 2.2. $\qquad\qquad\square$

**Proof of proposition 4.**

**Proposition 4** (maxent NE). *The game $\max_{\mathbf{p} \in \Delta_n} \min_{\mathbf{q} \in \Delta_n} \mathbf{p}^{\mathsf{T}}\mathbf{A}\mathbf{q}$, where $\mathbf{A}$ is antisymmetric, has a unique symmetric Nash equilibrium $(\mathbf{p}^*, \mathbf{p}^*)$, with greater entropy than any other Nash equilibrium.*

*Proof.* The Nash equilibria in a two-player zero-sum are rectangular: if $(\mathbf{p}, \mathbf{q})$ and $(\mathbf{u}, \mathbf{v})$ are Nash equilibria then so are $(\mathbf{p}, \mathbf{v})$ and $(\mathbf{u}, \mathbf{q})$. Further, they form a convex polytope. Since $\mathbf{A}^{\mathsf{T}} = -\mathbf{A}$, the set of Nash equilibria is also symmetric: if $(\mathbf{p}, \mathbf{q})$ is a Nash equilibrium then so is $(\mathbf{q}, \mathbf{p})$. The entropy $H(\mathbf{p}) := -\sum_{i=1}^n p_i \log p_i$ is strictly concave and therefore achieves a unique maximum on the compact, convex, symmetric set of Nash equilibria. $\qquad\qquad\square$

# C Proof of theorem 1

**Theorem** (main result for AvA). *The maxent NE has the following properties:*

P1. **Invariant:** *Nash averaging, with respect to the maxent NE, is invariant to redundancies in* $\mathbf{A}$.

P2. **Continuous:** *If* $\mathbf{p}^*$ *is a Nash for* $\hat{\mathbf{A}}$ *and* $\epsilon = \|\mathbf{A} - \hat{\mathbf{A}}\|_{max}$ *then* $\mathbf{p}^*$ *is an* $\epsilon$*-Nash for* $\mathbf{A}$.

P3. **Interpretable:** *(i) The maxent NE on* $\mathbf{A}$ *is the uniform distribution,* $\mathbf{p}^* = \frac{1}{n}\mathbf{1}$, *iff the meta-game is* cyclic, *i.e.* $\operatorname{div}(\mathbf{A}) = \mathbf{0}$. *(ii) If the meta-game is* transitive, *i.e.* $\mathbf{A} = \operatorname{grad}(\mathbf{r})$, *then the maxent NE is the uniform distribution on the player(s) with highest rating(s) – there could be a tie.*

## C.1 Proof of theorem 1.1

First, we more precisely formalize invariance to redundancy.

**Definition 3** (invariance to copying the last row and column). *Given antisymmetric matrix* $\mathbf{A}_{n \times n}$, *denote the right-most column by* $\mathbf{a}_n$. *Assume the right-most column (and bottom row) of* $\mathbf{A}$ *differs from all other columns. Construct antisymmetric matrix*

$$\mathbf{A}'_{(n+1)\times(n+1)} = \begin{pmatrix} \mathbf{A} & \mathbf{a}_n \\ -\mathbf{a}_n^\mathsf{T} & 0 \end{pmatrix} \tag{1}$$

*by adding an additional copy of the right-most column (and bottom row) to* $\mathbf{A}$. *A family of functions*

$$\left\{ \mathbf{p}_k : \{antisymmetric\ k \times k\ matrices\} \to \mathbb{R}^k \right\}_{k=1}^{\infty}$$

*is invariant to adding a row and column according to* (1) *if*

$$\mathbf{p}_{n+1}(\mathbf{A}')^\mathsf{T} = \left( \mathbf{p}_n(\mathbf{A})[1], \ldots, \mathbf{p}_n(\mathbf{A})[n-1], \frac{\mathbf{p}_n(\mathbf{A})[n]}{2}, \frac{\mathbf{p}_n(\mathbf{A})[n]}{2} \right).$$

If the copied row is not unique and receives positive mass under maxent Nash, then maxent Nash will already be spreading mass across the copies. In that case, adding *yet another* copy will result in the maxent Nash on the larger mass spreading mass evenly across all copies.

**Lemma 1.** *Suppose* $\mathbf{A}_{n \times n}$ *is antisymmetric with Nash equilibrium* $\mathbf{p}$. *Construct* $\mathbf{A}'$ *from* $\mathbf{A}$ *by adding a redundant copy of the right-most column and bottom row according* (1). *Then*

$$\mathbf{p}'_\alpha = \left( p_1, \ldots, p_{n-1}, \alpha \cdot p_n, (1-\alpha) \cdot p_n \right)$$

*is a Nash equilibrium for* $\mathbf{A}'$ *for all* $\alpha \in [0, 1]$. *Conversely, if* $\mathbf{p}'$ *is a Nash equilibrium for* $\mathbf{A}'$ *then*

$$(p'_1, \ldots, p'_{n-1}, p'_n + p'_{n+1})$$

*is a Nash equilibrium for* $\mathbf{A}$.

*Proof.* Since the value of the game on $\mathbf{A}$ is zero, it follows that $\mathbf{p}$ is a Nash equilibrium iff all the coordinates of $\mathbf{A}\mathbf{p}$ are nonnegative, i.e. $\mathbf{A}\mathbf{p} \succeq \mathbf{0}_{n \times 1}$. Direct computation shows that $\mathbf{A}'\mathbf{p}'_\alpha \succeq \mathbf{0}_{(n+1)\times 1}$, which implies $\mathbf{p}'_\alpha$ is a Nash equilibrium for $\mathbf{A}'$. The converse follows similarly. $\square$

Finally, we prove theorem 1.2.

*Proof.* The definition and lemma are stated in the particular case where the last column and row are copied. This is simply for notational convenience. They generalize trivially to copying arbitrary row/columns into arbitrary positions, and can be applied inductively to the cases where $\mathbf{A}'$ is constructed from $\mathbf{A}$ by inserting multiple redundant copies. $\square$

## C.2 Proof of theorem 1.2

Recall the max-norm on matrices is $\|\mathbf{S}\|_{\max} := \max_{ij} |\mathbf{S}_{ij}|$.

**Definition 4** ($\epsilon$-Nash equilibrium). *A joint strategy* $(\mathbf{p}^*, \mathbf{q}^*)$ *is an $\epsilon$-Nash equilibrium for* $\mathbf{A}$ *if the benefit from either player deviating, separately, is at most $\epsilon$:*

$$\max_{\mathbf{p}'}(\mathbf{p}' - \mathbf{p}^*)^{\mathsf{T}}\mathbf{A}\mathbf{q}^* \leq \epsilon \quad \text{and} \quad \max_{\mathbf{q}'}(\mathbf{p}^*)^{\mathsf{T}}\mathbf{A}(\mathbf{q}^* - \mathbf{q}') \leq \epsilon.$$

We are now ready to prove

**P2 Continuous:** If $\mathbf{p}^*$ is a Nash for $\hat{\mathbf{A}}$ and $\epsilon = \|\mathbf{A} - \hat{\mathbf{A}}\|_{\max}$ then $\mathbf{p}^*$ is an $\epsilon$-Nash for $\mathbf{A}$.

*Proof.* Suppose $(\mathbf{p}^*, \mathbf{p}^*)$ is a Nash equilibrium for the antisymmetric matrix $\hat{\mathbf{A}}$. Observe that

$$(\mathbf{p}' - \mathbf{p}^*)^{\mathsf{T}}\mathbf{A}\mathbf{p}^* = (\mathbf{p}')^{\mathsf{T}}\hat{\mathbf{A}}\mathbf{p}^* + (\mathbf{p}')^{\mathsf{T}}\mathbf{A}\mathbf{p}^* - (\mathbf{p}')^{\mathsf{T}}\hat{\mathbf{A}}\mathbf{p}^*$$

for any distribution $\mathbf{p}'$ because $(\mathbf{p}^*)^{\mathsf{T}}\mathbf{A}\mathbf{p}^* = 0$ since $\mathbf{A}$ is antisymmetric. It follows that

$$\max_{\mathbf{p}'}\left\{(\mathbf{p}' - \mathbf{p}^*)^{\mathsf{T}}\mathbf{A}\mathbf{p}^*\right\} \leq \max_{\mathbf{p}'}\left\{(\mathbf{p}')^{\mathsf{T}}\hat{\mathbf{A}}\mathbf{p}^*\right\} + \max_{\mathbf{p}'}\left\{(\mathbf{p}')^{\mathsf{T}}(\mathbf{A} - \hat{\mathbf{A}})\mathbf{p}^*\right\}$$

The first term on the right-hand-side is $\leq 0$ since $\mathbf{p}^*$ is a Nash equilibrium for $\hat{\mathbf{A}}$ and the value of the game is zero. The second term on the right-hand-side is $\leq \epsilon$ because $\|\mathbf{A} - \hat{\mathbf{A}}\|_{\max} \leq \epsilon$ and $\mathbf{p}'$ and $\mathbf{p}^*$ are probability distributions. $\square$

Note that since $\|\mathbf{S}\|_{\max} \leq \|\mathbf{S}\|_2 \leq \|\mathbf{S}\|_F$ for any $\mathbf{S}$, the divergence from Nash is also controlled by how well $\hat{\mathbf{A}}$ approximates $\mathbf{A}$ in the operator or Frobenius norms.

The proof is adapted from the proof of the following lemma in [58].

**Lemma 2** (Nash on approximate games). *Suppose* $(\mathbf{p}^*, \mathbf{q}^*)$ *is a Nash equilibrium for* $\hat{\mathbf{A}}$ *and that* $\epsilon = \|\mathbf{A} - \hat{\mathbf{A}}\|_{max}$. *Then* $(\mathbf{p}^*, \mathbf{q}^*)$ *is a $2\epsilon$-Nash for* $\mathbf{A}$.

Our result is slightly sharper because we specialize to antisymmetric matrices.

## C.3 Proof of theorem 1.3

*Proof.* **(i)** If $\mathrm{div}(\mathbf{A}) = \mathbf{0}$ then $\frac{1}{n}\mathbf{1}^{\mathsf{T}}\mathbf{A} = \mathbf{0}^{\mathsf{T}}$ and $\mathbf{A}\frac{1}{n}\mathbf{1} = \mathbf{0}$ implying the uniform distribution is a Nash equilibrium because there is no incentive to deviate from $(\frac{1}{n}\mathbf{1}, \frac{1}{n}\mathbf{1})$. The uniform distribution also has maximum entropy. Conversely, suppose $\mathrm{div}(\mathbf{A}) \neq \mathbf{0}$. Then $\mathbf{1}^{\mathsf{T}}\mathbf{A}$ has at least one positive and one negative coordinate because we know $\mathbf{1}^{\mathsf{T}}\mathbf{A}\mathbf{1} = 0$ by antisymmetry. It follows that if the row player chooses $\frac{1}{n}\mathbf{1}^{\mathsf{T}}$ then the column player is incentivized to choose a distribution with more mass on the positive coordinate and less on the negative. In other words, the column player will not play the uniform distribution, and the uniform distribution is therefore not a Nash equilibrium.

**(ii)** By assumption $\mathbf{A} = \mathrm{grad}(\mathbf{r}) = \mathbf{r}\mathbf{1}^{\mathsf{T}} - \mathbf{1}\mathbf{r}^{\mathsf{T}}$, so $\mathbf{p}^{\mathsf{T}}\mathbf{A}\mathbf{q} = \mathbf{p}^{\mathsf{T}}\mathbf{r} - \mathbf{r}^{\mathsf{T}}\mathbf{q}$ decouples into an independent maximization problem with respect to $\mathbf{p}$ and minimization problem with respect to $\mathbf{q}$. It follows that the optimal distribution $\mathbf{p}^*$ concentrates mass on the maximal coordinate(s) of $\mathbf{r}$ and so does $\mathbf{q}^*$. That is, to be a Nash equilibrium, $\mathbf{p}^*$ and $\mathbf{q}^*$ must place their mass on the maximal coordinate if it is unique and can distribute it arbitrarily over the set of maximal coordinates if there is a tie. Adding the condition that the Nash equilibrium has maximum entropy entails placing the uniform distribution over the maximal coordinate(s). $\square$

## D Nash averaging for agent-vs-task

Given score matrix $\mathbf{S}_{m \times n}$, construct antisymmetric matrix

$$\mathbf{A}_{(m+n) \times (m+n)} = \begin{pmatrix} \mathbf{0}_{m \times m} & \mathbf{S}_{m \times n} \\ -\mathbf{S}_{n \times m}^{\mathsf{T}} & \mathbf{0}_{n \times n} \end{pmatrix}.$$

Note this differs from the antisymmetrization used in section 3.3, see next remark.

*Remark* 1. The graph structure underlying AvT is *bipartite*: agents interact with tasks and tasks with agents, but there are no direct agent-agent or task-task interactions. When done in full generality, the definitions of $\mathrm{div}$, $\mathrm{grad}$ and $\mathrm{curl}$ take into account the graph structure, see [48]. In particular, $\mathrm{div}$, $\mathrm{grad}$ and $\mathrm{curl}$ are computed differently on bipartite graphs than fully connected graphs – note the definitions in section 2.2 are specific to fully connected graphs. Working in full generality is overkill for our purposes. It suffices to introduce slightly *ad hoc* notation to handle the specific case of AvT.

Introduce the notation

$$\mathrm{div}_a(\mathbf{S}) = \frac{1}{m}\mathbf{S} \cdot \mathbf{1}_{m \times 1} \quad \text{and} \quad \mathrm{div}_e(\mathbf{S}) = -\frac{1}{n}\mathbf{S}^{\mathsf{T}} \cdot \mathbf{1}_{n \times 1},$$

where $\mathrm{div}_a(\mathbf{S})$ measures uniform **average skill of agents** on tasks and $\mathrm{div}_e(\mathbf{S})$ measures uniform **average difficulty of tasks** for agents. Let

$$\mathrm{grad}(\mathbf{s}, \mathbf{d}) = \mathbf{s} \cdot \mathbf{1}^{\mathsf{T}} - \mathbf{1} \cdot \mathbf{d}^{\mathsf{T}}.$$

Define a two-player zero-sum meta-game

$$\max_{(\mathbf{p}_a, \mathbf{p}_e) \in \Delta_m \times \Delta_n} \min_{(\mathbf{q}_a, \mathbf{q}_e) \in \Delta_m \times \Delta_n} \left(\mathbf{p}_a; \mathbf{p}_e\right)^{\mathsf{T}} \mathbf{A}\left(\mathbf{q}_a; \mathbf{q}_e\right).$$

The setup is the same as for AvA in the main text except the row and column meta-players each play two distributions: one on agents and one on tasks/environments. The same argument as in proposition 4 shows there is a unique symmetric maxent Nash equilibrium $\left((\mathbf{p}_a^*, \mathbf{p}_e^*), (\mathbf{p}_a^*, \mathbf{p}_e^*)\right)$.

**Definition 5.** *The **maxent Nash evaluation method** for AvT is*

$$\mathcal{E}_m : \left\{evaluation\ data\right\} = \left\{antisymmetric\ matrices\right\} \xrightarrow{maxent\ NEs} \left[\left\{players\right\} \xrightarrow{Nash\ averages} \mathbb{R}\right],$$

*where $\mathbf{p}_a^*$ and $\mathbf{p}_e^*$ are the **maxent Nash equilibria** over agents and environments and $\mathbf{n}_e := -\mathbf{S}^{\mathsf{T}} \cdot \mathbf{p}_a^*$ and $\mathbf{n}_a := \mathbf{S} \cdot \mathbf{p}_e^*$ are the **Nash averages** quantifying difficulty of environments (Nash averaged over agents) and skill of agents (Nash averaged over environments) respectively.*

Suppose $\mathbf{S}$ decomposes as

$$\mathbf{S} = \mathrm{grad}(\mathbf{s}, \mathbf{d}) + \tilde{\mathbf{S}},$$

where $\mathrm{div}_a(\tilde{\mathbf{S}}) = \mathbf{0}$ and $\mathrm{div}_e(\tilde{\mathbf{S}}) = \mathbf{0}$. Observe that

$$\left(\mathbf{p}_a; \mathbf{p}_e\right)^{\mathsf{T}} \mathbf{A}\left(\mathbf{q}_a; \mathbf{q}_e\right) = \mathbf{s}^{\mathsf{T}}(\mathbf{p}_a - \mathbf{q}_a) + \mathbf{d}^{\mathsf{T}}(\mathbf{p}_e - \mathbf{q}_e) + \mathbf{p}_a^{\mathsf{T}}\tilde{\mathbf{S}}\mathbf{q}_e + \mathbf{q}_a^{\mathsf{T}}\tilde{\mathbf{S}}\mathbf{p}_e$$

If $\tilde{\mathbf{S}} \equiv \mathbf{0}_{m \times n}$ then the game reduces to

$$\left(\mathbf{p}_a; \mathbf{p}_e\right)^{\mathsf{T}} \mathbf{A}\left(\mathbf{q}_a; \mathbf{q}_e\right) = \mathbf{s}^{\mathsf{T}}(\mathbf{p}_a - \mathbf{q}_a) + \mathbf{d}^{\mathsf{T}}(\mathbf{p}_e - \mathbf{q}_e)$$

and so the row player maximizes its payoff by putting all its agent-mass on the most skillful agent(s) and all of its environment-mass on the most difficult task(s) – and similarly for the column player.

It is easy to check that the maxent Nash $(\mathbf{p}_a^*, \mathbf{p}_e^*)$ has $\mathbf{p}_a$ a uniform distribution on all agents iff $\mathrm{div}_a(\mathbf{S}) = \mathbf{0}$, and has $\mathbf{p}_e$ a uniform distribution on tasks iff $\mathrm{div}_e(\mathbf{S}) = \mathbf{0}$. We thus obtain

**Theorem 2** (main result for AvT). *The maxent NE has the following properties:*

*P1. **Invariant:** Nash averaging is invariant to redundancies in $\mathbf{A}$.*

*P2. **Continuous:** If $(\mathbf{p}_a^*, \mathbf{p}_e^*)$ is a Nash for $\hat{\mathbf{A}}$ and $\frac{\epsilon}{2} = \|\mathbf{A} - \hat{\mathbf{A}}\|_{max}$ then $(\mathbf{p}_a^*, \mathbf{p}_e^*)$ is an $\epsilon$-Nash for $\mathbf{A}$.*

*P3. **Interpretable:** (i) The agent component of the maxent NE on $\mathbf{A}$ is the uniform distribution on agents, $\mathbf{p}_a^* = \frac{1}{n}\mathbf{1}$, iff $\mathrm{div}_a(\mathbf{A}) = \mathbf{0}$.*
*(ii) The task component of the maxent NE on $\mathbf{A}$ is the uniform distribution on tasks, $\mathbf{p}_e^* = \frac{1}{m}\mathbf{1}$, iff $\mathrm{div}_e(\mathbf{A}) = \mathbf{0}$.*
*(iii) If the meta-game is* transitive, *i.e. $\mathbf{A} = \mathrm{grad}(\mathbf{s}, \mathbf{d})$, then the maxent NE is the uniform distribution on the most skillful agent(s) and the uniform distribution on the most difficult task(s) – there could be ties.*

Figure 3: **Visualizing Schur decompositions.** (A) Rows of $\mathbf{Q}_{4\times2}^{\mathbf{T}}$ form a straight line, reflecting the transitive structure of $\mathbf{T}$. (B): Rows of $\mathbf{Q}_{4\times2}^{\mathbf{C}}$ lie on a circle centered at the origin.

## E    Code for computing mElo$_2$ updates

The routine `mElo2_update` takes as input: a pair of players $i, j$, the probability `p_ij` of player `i` beating player `j` (which could be 0 or 1 if only a single match is observed on the given round), the rating vector `r` and the $n \times 2$ matrix `c` quantifying non-transitive interactions. It returns updates to the `i`th and `j`th entries of `r` and `c`.

```
def mElo2_update(i, j, p_ij, r, c):
  p_hat_ij = sigmoid(r[i] − r[j] + c[i, 0] ∗ c[j, 1] − c[j, 0] ∗ c[i, 1])
  delta = p_ij − p_hat_ij
  r_update = [16 ∗ delta, −16 ∗ delta]
    # r has higher learning rate than c
  c_update = [
    [+delta ∗ c[j, 1], −delta ∗ c[i, 1]],
    [−delta ∗ c[j, 0], +delta ∗ c[i, 0]]
  ]
  return r_update, c_update
```

## F    On the geometry of antisymmetric matrices

This section provides some intuition for multidimensional Elo ratings by describing some of the underling geometry.

### F.1    Visualizing the Schur decomposition

Consider the following logit matrices:

| $\mathbf{T}$ | $A$ | $B$ | $C$ | $D$ |
|---|---|---|---|---|
| $A$ | 0 | 1 | 2 | 3 |
| $B$ | -1 | 0 | 1 | 2 |
| $C$ | -2 | -1 | 0 | 1 |
| $D$ | -3 | -2 | -1 | 0 |

and

| $\mathbf{C}$ | $A$ | $B$ | $C$ | $D$ |
|---|---|---|---|---|
| $A$ | 0 | 1 | 0 | -1 |
| $B$ | -1 | 0 | 1 | 0 |
| $C$ | 0 | -1 | 0 | 1 |
| $D$ | 1 | 0 | -1 | 0 |

Note that $\mathbf{T}$ is transitive since $\mathbf{T} = \mathrm{grad} \circ \mathrm{div}(\mathbf{T})$ and $\mathbf{C}$ is cyclic since $\mathrm{div}(\mathbf{C}) = \mathbf{0}$. Both matrices have rank two, so the thin Schur decomposition can be written

$$\mathbf{Q}_{4\times2}\mathbf{\Lambda}_{2\times2}\mathbf{Q}_{2\times4}^{\mathsf{T}}$$

in either case. The matrices $\mathbf{Q}_{4\times2}^{\mathbf{T}}$ and $\mathbf{Q}_{4\times2}^{\mathbf{C}}$ arising from the respective Schur decompositions

$$\mathbf{Q}_{4\times2}^{\mathbf{T}} = \begin{pmatrix} 0.793 & 0.267 \\ 0.538 & -0.101 \\ 0.284 & -0.469 \\ 0.029 & -0.836 \end{pmatrix} \quad \text{and} \quad \mathbf{Q}_{4\times2}^{\mathbf{C}} = \begin{pmatrix} -0.296 & -0.642 \\ -0.642 & 0.296 \\ 0.296 & 0.642 \\ 0.642 & -0.296 \end{pmatrix}$$

Figure 4: **Visualizing logits.** The entry $\mathbf{A}_{ij}$ of $\mathbf{A}$ is $\lambda$ times the *signed area* of the parallelogram covered by the origin $(0,0)$, $\mathbf{Q}_{i,\bullet}$, $\mathbf{Q}_{i,\bullet} + \mathbf{Q}_{j,\bullet}$ and $\mathbf{Q}_{j,\bullet}$, where $\mathbf{Q}_{i,\bullet}$ and $\mathbf{Q}_{j,\bullet}$ are vectors corresponding to row $i$ and row $j$ of $\mathbf{Q}_{4\times 2}$.

can separately be thought of (and plotted) as four two-vectors, see figure 3. The rows of $\mathbf{Q}_{4\times 2}^{\mathbf{T}}$ form a straight line, see panel A. More generally, if $\mathbf{A} = \text{grad}(\mathbf{r})$ and $\mathbf{r}$ is not identically zero then $\mathbf{A}$ is rank two with thin Schur decomposition $\mathbf{A} = \mathbf{Q}_{n\times 2}\mathbf{\Lambda}_{2\times 2}\mathbf{Q}_{2\times n}^{\mathsf{T}}$, where the rows of $\mathbf{Q}$ are points along a line. More precisely, the rows are of the form

$$\mathbf{q}_k = \mathbf{a} + \alpha_k \mathbf{b} \quad \text{for } k \in \{1,\ldots,n\}$$

where $\mathbf{a}, \mathbf{b} \in \mathbb{R}^2$ and $\alpha_k \in \mathbb{R}$.

The rows of $\mathbf{Q}_{4\times 2}^{\mathbf{C}}$ lie on a circle centered at the origin, see panel B. This does not generalize to arbitrary cyclic matrices. However, the geometry of antisymmetric matrices has interesting connections with areas of parallelograms and the complex plane, see next subsection.

### F.2   Areas and phases

Observe that

$$(u_1 \quad u_2) \begin{pmatrix} 0 & \lambda \\ -\lambda & 0 \end{pmatrix} \begin{pmatrix} v_1 \\ v_2 \end{pmatrix} = \lambda \cdot (u_1 v_2 - u_2 v_1), \tag{2}$$

which is $\lambda$ times the signed area of the parallelogram covered by $(0,0)$, $\mathbf{u}$, $\mathbf{u} + \mathbf{v}$, and $\mathbf{v}$. It follows that the Schur decomposition breaks $\mathbb{R}^n$ into an orthogonal collection of two-dimensional spaces – one for each $\pm\lambda_i$ block – and that the entries $\mathbf{A}_{ij}$ of $\mathbf{A}$ are sums of signed areas of parallelograms, one parallelogram per two-dimensional subspace. The case where there is a single two-dimensional subspace (since the rank of $\mathbf{A}$ is two) is illustrated in figure 4.

Alternatively, introduce complex numbers $w = u_1 + iu_2$ and $z = v_1 + iv_2$, where $w = |w| \cdot e^{i\cdot\phi_w}$ and $z = |z| \cdot e^{i\cdot\phi_z}$ in polar coordinates. Then, (2) can be rewritten

$$(u_1 \quad u_2) \begin{pmatrix} 0 & \lambda \\ -\lambda & 0 \end{pmatrix} = \text{Im}(w \cdot \bar{z}) = |w| \cdot |z| \sin(\phi_w - \phi_z).$$

### F.3   Comment on Schur and Hodge

The Schur decomposition is not guaranteed to be compatible with the Hodge decomposition. That is, $\text{grad}(\mathbf{A})$ is not necessarily the span of two rows of the matrix $\mathbf{Q}_{n\times r}$ arising in the Schur decomposition. Roughly, this happens when $\tilde{\mathbf{A}} := \mathbf{A} - \text{grad} \circ \text{div}(\mathbf{A})$ is not a good $(\text{rank}_{\mathbf{A}} - 2)$-approximation to $\mathbf{A}$ in the $\|\cdot\|_2$-norm.

We recommend to first extract the transitive component and then perform the Schur decomposition on $\tilde{\mathbf{A}} = \text{rot}(\mathbf{A})$. Although this may not always be optimal with respect to the $\|\cdot\|_2$-norm, it has the important advantage that $\text{div}(\mathbf{A})$ is readily understood by humans as a measure of average performance.

|                  | Nash probability | Nash average | Uniform average |
|------------------|------------------|--------------|-----------------|
| breakout         | 0.000            | −0.450       | −0.457          |
| seaquest         | 0.000            | −0.487       | −0.298          |
| tutankham        | 0.000            | −0.798       | −0.619          |
| name_this_game   | 0.000            | −0.751       | −0.557          |
| robotank         | 0.000            | −0.689       | −0.687          |
| gopher           | 0.000            | −0.370       | −0.216          |
| space_invaders   | 0.000            | −0.387       | −0.218          |
| ms_pacman        | 0.000            | −0.850       | −0.500          |
| surround         | 0.000            | −0.807       | −0.453          |
| qbert            | 0.000            | −0.541       | −0.376          |
| bowling          | 0.000            | −0.553       | −0.328          |
| kangaroo         | 0.000            | −0.659       | −0.561          |
| krull            | 0.000            | −0.622       | −0.669          |
| yars_revenge     | 0.000            | −0.534       | −0.245          |
| solaris          | 0.000            | −0.552       | −0.249          |
| double_dunk      | 0.318            | −0.354       | −0.370          |
| skiing           | 0.000            | −0.576       | −0.461          |
| centipede        | 0.287            | −0.354       | −0.146          |
| time_pilot       | 0.000            | −0.624       | −0.514          |
| demon_attack     | 0.000            | −0.542       | −0.359          |
| asteroids        | 0.000            | −0.366       | −0.084          |
| berzerk          | 0.000            | −0.611       | −0.337          |
| crazy_climber    | 0.000            | −0.610       | −0.592          |
| freeway          | 0.000            | −0.952       | −0.729          |
| wizard_of_wor    | 0.000            | −0.456       | −0.327          |
| zaxxon           | 0.000            | −0.657       | −0.429          |
| ice_hockey       | 0.000            | −0.420       | −0.357          |
| road_runner      | 0.000            | −0.587       | −0.567          |
| fishing_derby    | 0.000            | −0.748       | −0.662          |
| alien            | 0.000            | −0.686       | −0.353          |
| defender         | 0.000            | −0.549       | −0.425          |
| private_eye      | 0.215            | −0.354       | −0.079          |
| gravitar         | 0.000            | −0.509       | −0.165          |
| beam_rider       | 0.000            | −0.524       | −0.377          |
| phoenix          | 0.000            | −0.365       | −0.185          |
| assault          | 0.000            | −0.532       | −0.336          |
| kung_fu_master   | 0.000            | −0.706       | −0.521          |
| enduro           | 0.000            | −0.731       | −0.531          |
| montezuma_revenge| 0.048            | −0.354       | −0.074          |
| video_pinball    | 0.000            | −0.431       | −0.360          |
| chopper_command  | 0.000            | −0.548       | −0.391          |
| boxing           | 0.000            | −0.705       | −0.740          |
| battle_zone      | 0.000            | −0.612       | −0.431          |
| bank_heist       | 0.000            | −0.652       | −0.521          |
| pitfall          | 0.000            | −0.356       | −0.090          |
| pong             | 0.000            | −0.946       | −0.894          |
| asterix          | 0.133            | −0.354       | −0.166          |
| atlantis         | 0.000            | −0.463       | −0.283          |
| star_gunner      | 0.000            | −0.410       | −0.363          |
| amidar           | 0.000            | −0.494       | −0.288          |
| frostbite        | 0.000            | −0.594       | −0.288          |
| hero             | 0.000            | −0.606       | −0.350          |
| venture          | 0.000            | −0.571       | −0.276          |
| tennis           | 0.000            | −0.514       | −0.440          |

Figure 5: Evaluation of environments.

|                          | Nash probability | Nash average | Uniform average |
|--------------------------|------------------|--------------|-----------------|
| DQN_(w/o_MC)             | 0.000            | 0.030        | 0.189           |
| DQN_(with_MC)            | 0.000            | 0.115        | 0.191           |
| DQN-PixelCNN_(w/o_MC)    | 0.000            | 0.022        | 0.161           |
| DQN-PixelCNN_(with_MC)   | 0.000            | 0.148        | 0.212           |
| DQN                      | 0.000            | 0.132        | 0.343           |
| A3C                      | 0.000            | 0.149        | 0.426           |
| DDQN                     | 0.000            | 0.244        | 0.556           |
| PriorDDQN                | 0.000            | 0.213        | 0.543           |
| DuelDDQN                 | 0.034            | 0.354        | 0.600           |
| DistribDQN               | 0.000            | 0.185        | 0.400           |
| NoisyDQN                 | 0.297            | 0.354        | 0.755           |
| Rainbow                  | 0.000            | 0.314        | 0.122           |
| RANDOM                   | 0.000            | 0.012        | 0.032           |
| HUMAN                    | 0.328            | 0.354        | 0.470           |
| DQN_                     | 0.000            | 0.132        | 0.343           |
| DDQN_                    | 0.000            | 0.149        | 0.426           |
| DUEL                     | 0.000            | 0.213        | 0.543           |
| PRIOR                    | 0.135            | 0.354        | 0.529           |
| PRIOR_DUEL               | 0.000            | 0.214        | 0.590           |
| PopArt                   | 0.206            | 0.354        | 0.457           |

Figure 6: Evaluation of agents. Note, there are redundancies since agents are taken from multiple papers; these are ignored by Nash averaging.