[Reviews · NeurIPS 2018]

Reviewer 1



This paper applies game theory (of Nash equilibria, though 2-player zero-sum games suffice for most of the arguments) to the problem of evaluating a number of agents on a slate of tasks (and/or against other agents). The evaluation methods presented have the important property of being robust to non-systematic addition of more agents and tasks. The paper casts the problem in a correct mathematical framework leveraging the Hodge decomposition of skew-symmetric matrices, and its generalization in combinatorial Hodge theory. This allows for a very illuminating and general view of the structure of evaluation, leading to a generalization of the Elo rating that eschews the transitivity assumption by embedding ratings in more dimensions, as is required in general, and a "Nash averaging" method which I think will be the more lasting contribution. The position taken by the paper on evaluation is also increasingly important at the current time, in my opinion. There are strong connections to the contextual bandits setting in the dueling case, in which feedback is limited to preferences among pairs of actions, which the learner chooses on the fly ([1] and related work). The pairwise preference/dominance matrix between the actions is skew-symmetric for the same reasons, and there is a general notion of a best action, which reduces to the naive one(s) when the matrix represents a transitive set of relations. This is very relevant work that the authors should be aware of, though it is unmentioned. The argument made in the conclusion uses the fact that there really is no intrinsic cost to including agents/tasks in this setting, contrasting with e.g. the bandit setting (where generalization is unavoidably hampered by state space size). It makes the authors' conclusions here much stronger than in that line of work; the bandit work also needs to contend with limited feedback, which is not a problem here. It should be fairly straightforward to make this much more openly usable, given that the input is only the dominance matrix and the algorithms are quite simple. It is disappointing to see this unmentioned apart from the Matlab code given for multidimensional Elo, and I would like to see this and the Nash averaging made available as painlessly as possible (e.g. a web tool) because of the unmistakably wide applicability. (In this vein there is a small concern about the solution's lack of sparsity e.g. when there are many tasks; there are clearly ways to tackle this in the authors' framework, for instance by regularizing with something other than entropy; there has been progress in optimal transport along similar lines in recent years.) One natural question that I did not find raised in this manuscript, but which could be answered using these tools, is: which tasks should be added to an existing evaluation suite to measure as-yet-unmeasured "finer-scale" differences between agents? The conclusion could be edited a bit; I appreciated the physics references, but found it incongruous to mention the speed of light when, closer to the main NIPS audience, there are plenty of applications of symmetries in ML via group theory (e.g. Kondor's monograph on the subject, or more broadly the success of convolutions for encoding spatial locality & translational invariances). line 34: "Our goal is to treat tasks and agents symmetrically." - this is not earlier motivated, and was a bit unclear to me at this point in the paper, should be clarified because it is quite central to motivating the main antisymmetry framework. Could be addressed by first motivating with Elo by moving that discussion from the preliminaries. line 132: "Sneak peek" [1] "Contextual Dueling Bandits", COLT 2015.

Reviewer 2



This paper discusses alg evaluation, mostly in the context of adversarial learning. I have done a bit of work in the former area, but am no an expert in the latter. The main contribution is the anti-symmetric decomposition that allows using Nash equilibria. The authors use this method to improve the ELO rating, I liked the trick in using the logit-space transformation. The authors do a nice analysis of paper-rock. They give a good argument why ELO does not work. The discussion on intransitiv abilities: two players is ok, but can you call something intransitive a ranking? 4.1 should be the highlight of the paper, but I found it very vague, a bit like trying to understand quicksort from its type. Prop 4 deserves more. I really missed some info on computing NE at this point. Yes, it's great to have detail in the extra sections, but they should be an appendix Last, its important to have lofty goals, but the paper starts by claiming to address Q1-Q3. I would suggest either focusing the Intro on the task, or connecting back to the qs. In general, very nice work, but do consider discussing more your contribution and avoid sidetracks at the speed of light.

Reviewer 3



The paper proposes a new way to evaluate algorithms given their performance either on tasks or in a match against each other. Many papers evaluate new algorithms and tasks in ad-hoc ways, and it is often unclear which algorithm is better than which, and under what circumstances. The authors bring an educated approach to this issue by borrowing techniques from game theory. They consider a meta-game where one player chooses a team / distribution of algorithm and the other a team / distribution of tasks and propose ranking methods based on this game. There are two somewhat different approaches given in the paper. The first describes a decomposition of the algorithm-vs-task (AvT) or algorithm-vs-algorithm (AvA) matrix that lead to a low-rank approximation of sorts, giving ‘latent vectors’ for each algorithm that act as a multi-dimensional ranking. These provide a better representation of the algorithms than a one-dimensional ranking such as Elo, at the cost of being less interpretable. In particular they do not force a total order. The second method provides a ranking that tries to maintain a set of properties that seem very reasonable. This method provides each algorithm a score based on its comparison to the min-max strategy in the above mentioned game. The ranking is done according to the score. Summary: This is somewhat of a non-standard paper as it doesn’t bring any new technique or develop a new mathematical tool. Rather it adapts known techniques in game theory and attempts to bring order to a somewhat chaotic area of algorithm evaluation. I suspect that the decision of accepting / rejecting the paper could be somewhat subjective but my opinion is to accept the paper. Writing: The paper is a pleasure to read. The authors take time to explain their approach and the intuition for their decisions is presented clearly. Related literature: The authors should relate their work to the field of dueling bandits. The dueling bandit problem, motivated by IR where algorithms are compared against each other has very similar issue. It is a common case there that the preference between pairs is not transitive meaning alg1 > alg2, alg2 > alg3, alg3 > alg1. Despite that, the bandit problem requires a notion of the best algorithm and of regret, meaning a sub optimality measure for a non-optimal algorithm. The issue is typically solved there by assuming the existence of a Condorcet winner, meaning an algorithm that is likely to beat any other algorithm in a match w.p. larger than 1/2. Given that, the regret associated with a sub-optimal algorithm is determined by how likely it is to lose to this winner. This definition exactly matches the Nash averaging score given here. In fact, there is a paper for dueling bandits titled “Instance-dependent Regret Bounds for Dueling Bandits” that discusses the case where there is no Condorcet winner, and they propose a notion of regret that is strongly related to the Nash averaging score. Comments: * Equation at 157: Why are the Eigenvalues assumed to be approximately -i,i? Could you add an explanation? * Line 165: The norm of P is not one - but depends on the number of players. It would be more helpful to discuss something normalized like |P-P’|/|P|. Also, the objective is not to reconstruct P in the Frobenius norm but with logistic loss. Is it possible to present that instead? Finally, why is there a ‘slight’ improvement? 0.85 to 0.35 seems like a pretty big improvement.. * line 168: Add one line to describe what the algorithms are meant to solve * Is it true that n_A(i) is the payoff of algorithm i against the maxent NE strategy? This could be mentioned explicitly to demonstrate interpretability * Theorem 1: The proposed score is the Nash Averaging n_A, but the theorem gives properties of p^*. I understand that the proofs go through p^* but shouldn’t the main statement be written in terms of n_A? Also, the notation \eps-Nash should be defined before used in a theorem. * Normalizing scores (in AvT) seem like an important issue that is somewhat avoided - is there any educated way of doing that? Perhaps converting to quantiles so outliers will not be an issue?